

# Actuator behaviour modelling in IoT-Fog-Cloud simulation

Andras Markus[1], Mate Biro[1], Gabor Kecskemeti[2] and Attila Kertesz[1]

[1] Software Engineering Department, University of Szeged, Szeged, Hungary
[2] Institute of Information Technology, University of Miskolc, Miskolc, Hungary

## ABSTRACT

The inevitable evolution of information technology has led to the creation of IoT-Fog-Cloud systems, which combine the Internet of Things (IoT), Cloud Computing and Fog Computing. IoT systems are composed of possibly up to billions of smart devices, sensors and actuators connected through the Internet, and these components continuously generate large amounts of data. Cloud and fog services assist the data processing and storage needs of IoT devices. The behaviour of these devices can change dynamically (*e.g.* properties of data generation or device states). We refer to systems allowing behavioural changes in physical position (*i.e.* geolocation), as the Internet of Mobile Things (IoMT). The investigation and detailed analysis of such complex systems can be fostered by simulation solutions. The currently available, related simulation tools are lacking a generic actuator model including mobility management. In this paper, we present an extension of the DISSECT-CF-Fog simulator to support the analysis of arbitrary actuator events and mobility capabilities of IoT devices in IoT-Fog-Cloud systems. The main contributions of our work are: (i) a generic actuator model and its implementation in DISSECT-CF-Fog, and (ii) the evaluation of its use through logistics and healthcare scenarios. Our results show that we can successfully model IoMT systems and behavioural changes of actuators in IoT-Fog-Cloud systems in general, and analyse their management issues in terms of usage cost and execution time.

## INTRODUCTION

The Internet of Things (IoT) is estimated to reach over 75 billion smart devices around the world by 2025 (*Taylor, Baron & Schmidt, 2015*), which will dramatically increase the network traffic and the amount of data generated by them. IoT systems often rely on Cloud Computing solutions, because of its ubiquitous and theoretically infinite, elastic computing and storage resources. Fog Computing is derived from Cloud Computing to resolve the problems of increased latency, high density of smart devices and the overloaded communication channels, which also known as the bottleneck-effect.

Real-time IoT applications (*Ranjan et al., 2020*) require faster and more reliable data storage and processing than general ones, especially when data privacy is also a concern. The proximity of Fog Computing nodes to end users usually ensures short latency values, however these nodes are resource-constrained as well. Fog Computing can aid cloud nodes by introducing additional layers between the cloud and the IoT devices, where

Corresponding author
Andras Markus,
markusa@inf.u-szeged.hu

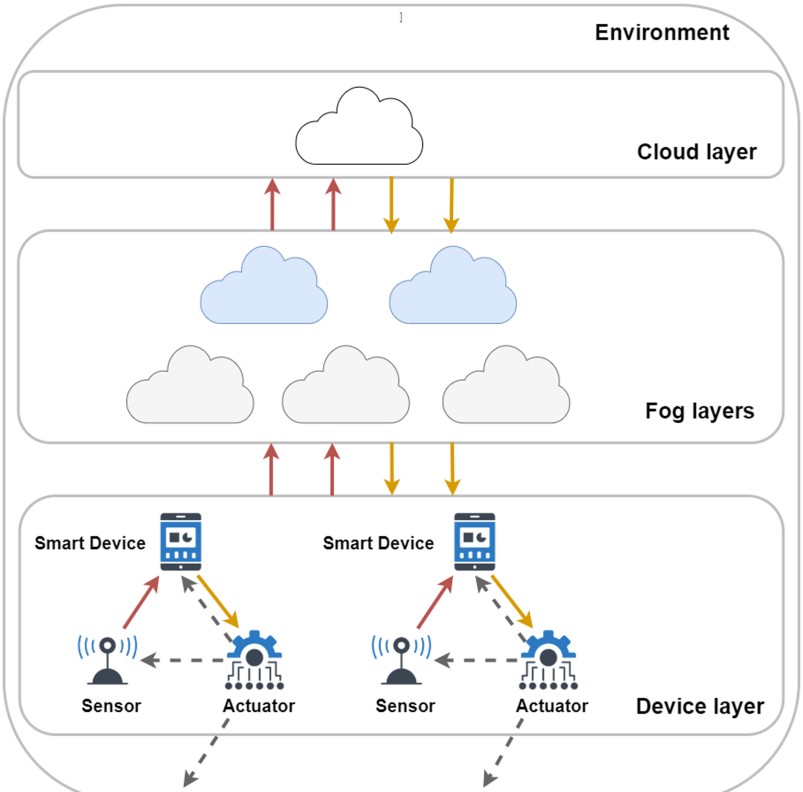

**Figure 1 The connections and layers of a typical fog topology.**

a certain part of the generated data can be processed faster (*Mahmud, Kotagiri & Buyya, 2018*).

A typical fog topology is shown in Fig. 1, where sensors and actuators of IoT devices are located at the lowest layer. Based on their configuration and type, things produce raw sensor data. These are then stored and processed on cloud and fog nodes (this data flow is denoted by red dotted arrows). Sensors are mostly resource-constrained and passive entities with restricted network connection, on the other hand, actuators ensure broad functionality with Internet connection and enhanced resource capacity (*Ngai, Lyu & Liu, 2006*).

They aspire to make various types of decisions by assessing the processed data retrieved from the nodes. This data retrieval is marked by solid orange arrows in Fig. 1. These actions can affect on the physical environment or refine the configuration of the sensors, such modification can be the increasing or decreasing of the data sampling period or extending the sampling period, this later results in different amounts of generated data. Furthermore, the embedded actuators can manipulate the behaviour of smart devices, for instance, restart or shutdown a device, and motion-related responses can also be expected. These kind of actions are noted by grey dashed arrows in Fig. 1.

In the surrounding world of IoT devices, location is often fixed; however, the Quality of Service (QoS) of these systems should also be provided at the same level in case of dynamic

and moving devices. Systems composed of IoT devices supporting mobility features are also known as the Internet of Mobile Things (IoMT) (*Nahrstedt et al., 2020*). Mobility can have a negative effect on the QoS to be ensured by fog systems, for instance, they could increase the delay between the device and the actual node it is connected to. Furthermore, using purely cloud services can limit the support for mobility (*Pisani et al., 2020*).

Wireless Sensor Networks (WSN) are considered as predecessors of the Internet of Things. In a WSN, the naming convention of sensor and actuator components follows publisher/subscriber or producer/consumer notions (*Sheltami, Al-Roubaiey & Mahmoud, 2016*), however IoT sensor and actuator appellations are commonly accepted by the IoT simulation community as well. Publishers (*i.e.* sensors or producers) share the data which are sensed in the environment, until then subscribers (*i.e.* actuators or consumers) react to the sensor data (or to an incoming message) with an appropriate action. In certain situations, actuators can have both of these roles, and behave as a publisher, especially when the result of a command executed by an actuator needs to be sent and further processed.

Investigating IoT-Fog-Cloud topologies and systems in real word is rarely feasible on the necessary scales, thus different simulation environments are utilised by researchers and system architects for such purpose. It can be observed that only a few of the currently available simulation tools deal with a minimal ability to model actuator and/or mobility events, which strongly restricts their usability. It implies that a comprehensive simulation solution, with an extendable, well-detailed mobility and actuator model, is missing for fog-enhanced IoT systems.

To address this open issue, we propose a generic actuator model for IoT-Fog-Cloud simulators and implement it by extending the DISSECT-CF-Fog (*Markus, Gacsi & Kertesz, 2020*) open-source simulator, to be able to model actuator components and mobility behaviour of IoT devices. As the main contributions of our work, our proposal enables: (i) more realistic and dynamic IoT behaviour modelling, which can be configured by using the actuator interface of IoT devices, (ii) the ability of representing and managing IoT device movement (IoMT), and (iii) the analysis of different types of IoT applications having actuator components in IoT-Fog-Cloud environments. Finally, the modelling of such complex systems are demonstrated through a logistics and a healthcare scenario.

The rest of the paper is structured as follows: "Related Work" introduces and compares the related works, "The Actuator and Mobility Models of Dissect-CF-Fog" presents our proposed actuator model and simulator extension. "Evaluation" presents our evaluation scenarios, and finally "Conclusion" concludes our work.

## RELATED WORK

According to the definition by *Bonomi et al. (2012)*, an actuator is a less limited entity than a sensor in terms of its network connectivity and computation power, since it is responsible for controlling or taking actions in an IoT environment. Usually actuators are identified as linear, motors, relays or solenoids to induce motion of a corresponding entity. The work in *Motlagh et al. (2020)* categorises actuators based on their energy source as following: (i) pneumatic, (ii) hydraulic, (iii) electric and finally (iv) thermal actuator,

however this kind of classification might restrict the usability of actuators to the energy sector.

The presence of actuators plays a vital role in higher level software tools for IoT as well, for instance in FogFlow (*Cheng et al., 2018*). It is an execution framework dedicated for service orchestrations over cloud and fog systems. This tool helps infrastructure operators to handle dynamic workloads of real IoT services enabling low latency on distributed resources. According to their definition, actuators perform actions (*e.g.* turning on/off the light) in an IoT environment, which can be coordinated by an external application.

The already existing, realised actuator solutions are well-known and commonly used in technical informatics; however the modelling of an actuator entity in simulation environments is not straightforward, and most of the simulation tools simply omit or simplify it, nevertheless actuators are considered as essential components of the IoT world.

Concerning IoT and fog simulation, a survey paper by *Svorobej et al. (2019)* compares seven simulation tools supporting infrastructure and network modelling, mobility, scalability, resource and application management. Unfortunately, in some cases the comparison is restricted to a binary decision, for instance if the simulator has a mobility component or not. Another survey by *Markus & Kertesz (2020)* examined 44 IoT-Fog-Cloud simulators, in order to determine the characteristics of these tools. A total of 11 parameters were used for the comparison, such as type of the simulator, the core simulator, publication date, architecture, sensor, cost, energy and network model, geolocation, VM management and lastly, source code metrics. These survey papers represent the starting point for our further investigations in the direction of geolocation and actuator modelling.

FogTorchPI (*Brogi, Forti & Ibrahim, 2018*) is a widely used simulator, which focuses on application deployment in fog systems, but it limits the possibilities of actuator interactions. *Tychalas & Karatza (2018)* proposed a simulation approach focusing on the cooperation of smartphones and fog, however the actuator component was not considered for the evaluation.

The CloudSim-based iFogSim simulator (*Gupta et al., 2016*) is one of the leading fog simulators within the research community, which follows the sense-process-actuate model. The actuator is declared as the responsible entity for the system or a mechanism, and the actualisation event is triggered when a task, which known as a Tuple, determining a certain amount of instruction and size in bytes, is received by the actuator. In the current implementation of iFogSim, this action has no significant effect, however custom events also can be defined by overriding the corresponding method, nevertheless no such events are created by default. The actuator component is determined by its connection and network latency. The original version of iFogSim does not support mobility, however the static, geographical location of a node is stored.

Another CloudSim extension is the EdgeCloudSim (*Sonmez, Ozgovde & Ersoy, 2018*), which aims to ensure mobility support in simulation environments. It associates the position information of a mobile device to a two-dimensional coordinate point, which can be updated dynamically. This simulation solution considers the nomadic mobility model, by its definition, a group of nodes moves randomly from one position to another. This

work also takes into account the attractiveness of a position to define the duration of stay at some place. Further mobility models can be created by extending the default class for mobility, but there is no actuator entity implemented in this approach.

The FogNetSim++ (*Qayyum et al., 2018*) can be used to model fog networks supporting heterogeneous devices, resource scheduling and mobility. In this paper six mobility strategies were proposed, and new mobility policies can also be added. This simulator aids the entity mobility models, which handles the nodes independently, and takes into account parameters such as speed, acceleration, direction in a three-dimensional coordinate system. Unfortunately, the source code of the simulator presents examples of the linear and circular mobility behaviour only. This simulation tool used no actuator model.

YAFS (*Lera, Guerrero & Juiz, 2019*) is a simulator to analyse IoT application deployments and mobile IoT scenarios. The actuator in this realisation is defined as an entity, which receives messages with the given number of instructions and bytes, similarly to the solution of iFogSim. The paper also mentioned dynamic user mobility, which takes into account different routes using GPX formats (it is used by application to depict data on the map), but this behaviour was not explained or experimented with.

*Jha et al. (2020)* proposed the IoTSim-Edge simulation framework by extending the CloudSim to model towards IoT and Edge systems. This simulator focuses on resource provisioning for IoT applications considering the mobility function and battery-usage of IoT devices, and different communication and messaging protocols as well. The IoTSim-Edge contains no dedicated class for the actuator components, nevertheless the representative class of an IoT device has a method for actuator events, which can be also overridden. There is only one predefined actuator event affecting the battery of an IoT device, however it was not considered during the evaluation phase by the authors. This simulation tool also takes into consideration the mobility of smart devices. The location of a device is represented by a three-dimensional coordinate system. Motion is influenced by a given velocity and range, where the corresponding device can move, and only horizontal movements are considered within the range by the default moving policy.

MobFogSim (*Puliafito et al., 2020*) aims to model user mobility and service migration, and it is one of the latest extension of the iFogSim, where actuators are supported by default. Furthermore, the actuator model was revised and improved to handle migration decisions, because migration is often affected by end user motions. To represent mobility, it uses a two-dimensional coordinate system, the users' direction and velocity. The authors considered real datasets as mobility patterns, which describe buses and routes of public transportation.

The comparison of related simulation based approaches is shown in Table 1. It highlights the existence of actuator and mobility interfaces, the base simulator of the approach and the programming language, in which the actual tool was written. We also denoted the year, when the simulation solution was released or published. It also reveals the leading trends for fog simulation. Based on *Markus & Kertesz (2020)*, more than 70% of the simulators are written in Java programming language and only 20% of them are developed using Python or C++. The rest of them are more complex applications (*i.e.* Android-based software). This survey also points out that mostly the network type of

**Table 1  Comparison of the related simulation tools.**

| Simulator | Actuator | Mobility | Core simulator | Prog. language | Year |
|---|---|---|---|---|---|
| DISSECT-CF-Fog (this work) | X | X | DISSECT-CF | Java | 2020 |
| iFogSim | X | – | CloudSim | Java | 2017 |
| EdgeCloudSim | – | X | CloudSim | Java | 2017 |
| FogNetSim++ | – | X | OMNet++ | C++ | 2018 |
| IoTSim-Edge | X | X | CloudSim | Java | 2019 |
| YAFS | X | X | – | Python | 2019 |
| MobFogSim | X | X | iFogSim | Java | 2020 |

**Table 2  Detailed characteristics of the related simulation tools.**

| Simulator | Communication direction | Actuator events | Mobility | Position |
|---|---|---|---|---|
| DISSECT-CF-Fog (this work) | • Sensor → Fog/Cloud → Actuator<br>• Sensor → Actuator | • 10 different predefined actions for actuation<br>• Adding new by overriding | • Nomadic<br>• Random Walk | • Latitude, Longitude |
| iFogSim | • Sensor → Fog → Actuator | • Default, but it can be overridden | – | • Coordinates |
| EdgeCloudSim | – | – | • Nomadic | • Coordinates |
| FogNetSim++ | – | – | • Linear<br>• Circular | • Coordinates |
| IoTSim-Edge | • Sensor → Fog Device → Actuator | • Default, but it can be overridden | • Linear | • Coordinates |
| YAFS | • Sensor → Service → Actuator | – | • Real dataset | • Latitude, Longitude |
| MobFogSim | • Mobile Sensor → Mobile Device → Mobile Actuator | • Migration | • Linear<br>• Real dataset | • Coordinates |

simulators is written in C++, which focuses on fine-grained network model, however these tools typically do not have predefined models and components for representing cloud and fog nodes, and VM management operations. The event-driven general purpose simulators are usually implemented in Java.

The actuator and mobility abilities of these simulators are further detailed in Table 2. The second column shows possible directions for transferring the sensor data (usually in the form of messages), in case the actuator interface is realised in the corresponding simulator. It can be observed that it basically follows similar logic in all cases. The third column highlights actuator events that can be triggered in a simulator. The fourth column shows the supported mobility options (we only listed the ones offered in their source code) and finally we denoted the position representation manner in the last column.

One can observe that there is a significant connection between mobility support and actuator functions, but only half of the investigated simulators applied both of them. Since the actuator has no commonly used software model within the latest simulation tools, developers omit it, or it is left to the users to implement it, which can be time consuming

(considering the need for additional validation). In a few cases, both actuator and mobility models are simplified or just rudimentary handled, thus realistic simulations cannot be performed.

In this paper, we introduce an actuator interface and mobility functionality for the DISSECT-CF-Fog simulator. We define numerous actuator events and mobility patterns to enhance and refine the actuator model of a simulated IoT system. To the best of our knowledge, no other simulation solution offers such enriched ways to model actuator components.

## THE ACTUATOR AND MOBILITY MODELS OF DISSECT-CF-FOG

The heterogeneity of interconnected IoT devices often raises difficulties in simulator solutions, as the creation of a model that comprehensively depicts the behaviour of these diverse components is challenging. In a simulation environment, a concrete type of any device is described by its characteristics. For instance, it does not really matter, if a physical machine utilises an AMD or an Intel processor, because the behaviour of the processor are modelled by the number of CPU cores and the processing power of one core, which should be defined in a realistic way. Following this logic, the actual realisation of an actuator entity—which follows the traditional subscriber model –, can be any type of actuator (*e.g.* motors or relays), if the effects of it are appropriately and realistically modelled. This means that in our actuator implementation, a command received by an actuator must affect the network load considering bandwidth and latency, moreover based on certain decisions the actuator should indicate changes in the behaviour of the IoT device or sensor (*e.g.* increasing data sensing frequency or changing the actual position). In case of IoMT, the traditional WSN model cannot be followed, hence moving devices can act as a publisher (monitoring) and subscriber as well (receiving commands related to movements).

Our proposed actuator interface of the DISSECT-CF-Fog simulator aims to provide a generic, unified, compact and platform-independent representation of IoT actuator components. DISSECT-CF-Fog is based on DISSECT-CF (*Kecskemeti, 2015*), which was proposed as a general purpose simulator to investigate the energy consumption of cloud infrastructures. The evolution phases of DISSECT-CF-Fog can be seen in Fig. 2, where each background colour represents a milestone of the development, and it also depicts the layers of the simulation tool.

Typical event-driven simulators are lacking predefined models for complex behaviours (*e.g.* considering both detailed network and computational resource utilisation), nevertheless DISSECT-CF has such abilities. It utilises its own discrete event simulation (DES) engine, which is responsible to manage the time-dependent entities (*Event System*) and also considers low-level computing resource sharing, for instance balancing network bandwidth (*Unified Resource Sharing*) or enabling the measurement of different energy usage patterns of resources (*Energy Modelling*). Through the *Infrastructure Simulation* and *Infrastructure Management* layers, general IaaS clouds can be modelled with different scheduling policies.

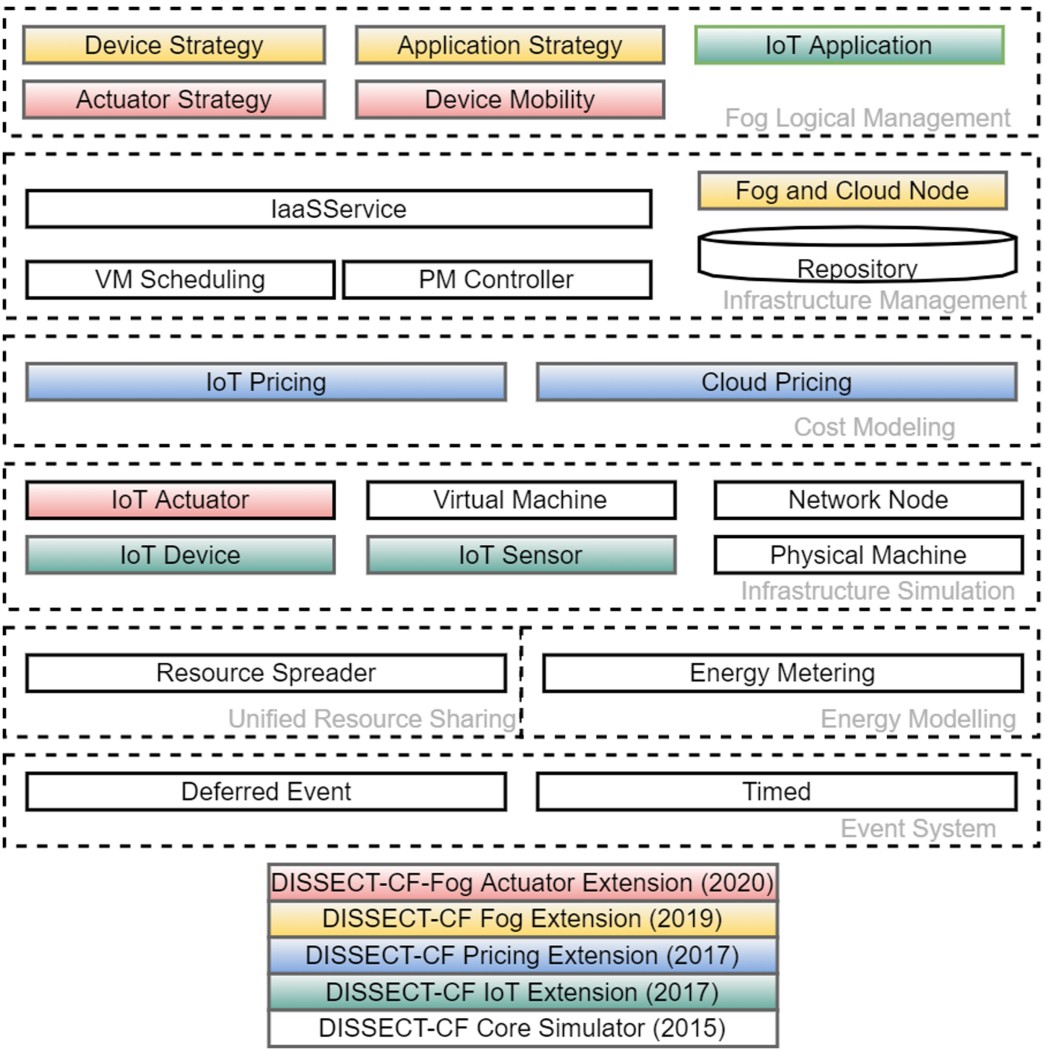

**Figure 2 The evolution of DISSECT-CF-Fog through its components.**

    The current version of DISSECT-CF-Fog is strongly built on the subsystems of the basic simulator, which is proven to be accurate. This system has been leveraged since 2017 to realise different aspects of complex IoT-Fog-Cloud systems. First, we added the typical components of IoT systems (denoted by green in Fig. 2) like *IoT Sensor*, *IoT Device* and *IoT Application*, to model various IoT use cases with detailed configuration options. The naming DISSECT-CF-Fog was introduced at the end of 2019, after developing the *Cost Modelling* layer to apply arbitrary IoT and cloud side cost schemes of any providers (shown by blue coloured boxes in Fig. 2). The tree main components of the fog extension, denoted by yellow (Fig. 2), are the *Fog and Cloud Node*, which are responsible for the creation of multi-tier fog topology, the *Device Strategy*, which chooses the optimal node for a device, and (iii) the *Application Strategy*, which enables offloading decisions between the entities of the fog topology. The strategies can take into account various parameters of the system, such as network properties (*e.g.* latency), cost and utilised CPU and memory.

The main contribution of this paper are denoted by red in Fig. 2. To satisfy the increasing need for a well-detailed and versatile simulator, we complete the IoT layer by adding the *IoT Actuator* component, with its corresponding management elements *Actuator Strategy* and *Device Mobility*, to realise the business logic for such related behaviours. In the former versions of DISSECT-CF-Fog, the position of IoT devices were static and fixed, and also the backward communication channels (from the computational nodes to actuators through the IoT devices) did not exist.

## Actuator model

In the layered architecture of IoT, actuators are located in the perception layer, which is often referred to as the lowest or physical layer that requires the most detailed level of abstraction in IoT.

In this paper, we are focusing mainly on software-based actuator solutions due to their increasing prevalence in the field of IoT. The DISSECT-CF-Fog actuator model is fairly abstract, hence it mainly focuses on the actuators' core functionality and its effect on the simulation results, but it does not go deep into specific actuator-device attributes.

The actuator interface should facilitate a more dynamic device layer and a volatile environment in a simulation. Therefore, it is preferred to be able to implement actuator components in any kind of simulation scenario, if needed. In our model, one actuator is connected to one IoT device for two reasons in particular: (i) it is observing the environment of the smart device and can act based on previously specified conditions, or (ii) it can influence some low-level sensor behaviour, for instance it changes the sampling interval of a sensor, resets or completely stops the smart device.

The latter indirectly conveys the conception of a reinterpreted actuator functionality for simulator solutions. The DISSECT-CF-Fog actuator can also behave as a low-level software component for sensor devices, which makes the model compound.

The actuator model of DISSECT-CF-Fog can only operate with compact, well-defined events, that specify the exact influence on the environment or the sensor. The set of predefined events during a simulation provides a restriction to the capability of the actuator and limits its scope to certain actions that are created by the user or already exist in the simulator. A brief illustration of sensor-based events are shown in Fig. 3.

The determination of the exact event, executed by the actuator, happens in a separate, reusable and extendable logic component. This logic component can serve as an actual actuator configuration, but can also be used as a descriptor for environmental changes and their relations to specific actuator events. This characteristic makes the actuator interface thoroughly flexible and adds some more realistic factors to certain simulation scenarios.

With the help of the logic component, the actuator interface works in an automatic manner. After a cloud or fog node has processed the data generated by the sensors, it sends a response message back to the actuator, which chooses an action to be executed. This models the typical sensor-service-actuator communication direction.

Unexpected actions may occur in real-life environments, which are hard to be defined by algorithms, and the execution of some events may not require cloud or fog processes, *e.g.* when a sensor fails. To be able to handle such issues, the actuator component is

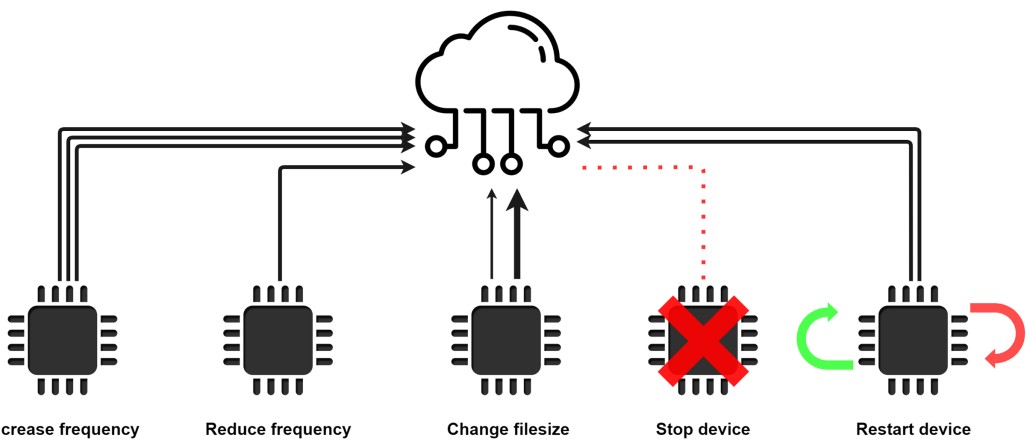

| Increase frequency | Reduce frequency | Change filesize | Stop device | Restart device |

**Figure 3  Low-level sensor events.**               

capable of executing events apart from its predefined configuration. This feature facilitates the immediate and direct communication between sensors and actuators.

For the proper behaviour of the actuator, the data representation in the simulator needs to be more detailed and comprehensive. Consequently, this extension of the DISSECT-CF-Fog simulator introduces a new type of data fragment in the system, to store specific details throughout the life-cycle of the sensor-generated data.

Finally, the DISSECT-CF-Fog actuator should be optional for simulation scenarios. In consideration of certain scenarios, where the examined results do not depend on the existence of actuator behaviours, the simulation can be run without the actuator component. This might significantly decrease the actual runtime of the simulation, as there could potentially be some computing heavy side effects, when applying actuator functionalities.

## Requirements for modelling the internet of mobile things

The proximity of computing nodes is the main principle of Fog Computing and it has numerous benefits, but mobile IoT devices may violate this criterion. These devices can move further away from their processing units, causing higher and unpredictable latency. When a mobile device moves out of range of the currently connected fog node, a new, suitable fog node must be provided. Otherwise, the quality of service would drastically deteriorate and due to the increased latencies, the fog and cloud nodes would hardly be distinguishable in this regard, resulting in losing the benefits of Fog Computing.

Another possible problem that comes with mobile devices is service migration. The service migration problem can be considered as when, where and how (W2H) questions. Service migration usually happens between two computing nodes, but if there is no fog node in an acceptable range, the service could be migrated to the smart device itself, causing lower performance and shorter battery time. However, service migration only makes sense when there are stateful services, furthermore it is beyond the topic of this paper, we consider stateless services and decisions of their transfer among the nodes only.

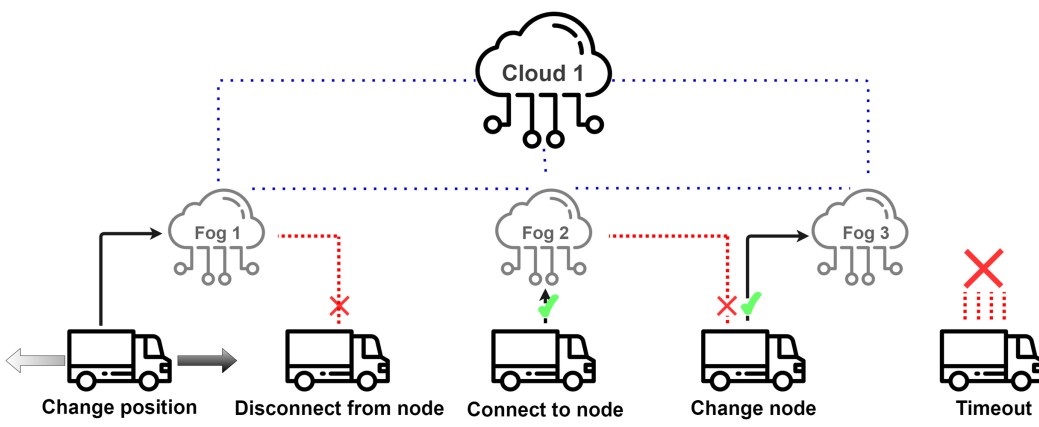

Figure 4 Actuator events related to mobility behaviour.

The physical location of fog nodes in a mobile environment is a major concern. Placing Fog Computing nodes too far from each other will result in higher latency or connection problems. In this case, IoT devices are unable to forward their data, hence they are never processed. Some devices may store their data temporarily, until they connect to a fog node, but this contradicts real-time data processing promises of fogs.

A slightly better approach would be to install fog nodes fairly dense in space to avoid the problem discussed above. However, there might be some unnecessary nodes in the system, causing a surplus in the infrastructure, which results in resource wastage.

Considering different mobility models for mobile networks in simulation environments have been researched for a while. The survey by *Camp, Boleng & Davies, 2002* presents seven entity and six group mobility models in order to replace trace files, which can be considered as the footprints of movements in the real world. Applying mobility models is a reasonable decision, because they mimic the movements of IoT devices in a realistic way. The advent of IoT and the technological revolution of smartphones have brought the need for seamless and real time services, which may require an appropriate simulation tool to develop and test the cooperation of Fog Computing and moving mobile devices.

The current extension of the DISSECT-CF-Fog was designed to create a precise geographical position representation of computing nodes (fog, cloud) and mobile devices and simulate the movements of devices based on specified mobility policies. As the continuous movement of these devices could cause connection problems we consider the following events shown in Fig. 4.

Examining the occurrence of these specific events can help in optimising the physical allocation of fog nodes depending on the mobility features of IoT devices.

## Actuator implementation in DISSECT-CF-Fog

DISSECT-CF-Fog is a discrete event simulator, which means there are dedicated moments, when system variables can be accessible and modifiable. The extended classes of the timing events, which can be recurring and deferred, ensure to create the dedicated time-dependent entities in the system.

As mentioned in "Actuator Model", a complex, detailed data representation in the simulator is mandatory in order to provide sufficient information for the actuator component. Data fragments are represented by *DataCapsule* objects in the system of DISSECT-CF-Fog. The sensor-generated data is wrapped in a well-parameterized *DataCapsule* object, and forwarded to an IoT *Application* located in a fog or cloud node to be processed. A *DataCapsule* object uses the following attributes:

- *source*: Holds a reference to the IoT device generating sensor data, so the system keeps track of the data source.
- *destination*: Holds a reference to the *Application* of a fog node where the data has been originally forwarded to.
- *dataFlowPath*: In some cases fog nodes cannot process the current data fragment, therefore they might send it to another one. This parameter keeps track of the visited fog nodes by the data before it has been processed.
- *bulkStorageObject*: Contains one or more sensor-generated data that has been wrapped into one *DataCapsule*.
- *evenSize*: The size of the response message sent from a fog node to the actuator component (in bytes). This helps to simulate network usage while sending information back to the actuator.
- *actuationNeeded*: Not every message from the IoT device requires an actuator response event. This logical value (true–false) holds true, if the actuator should take action after the data has been processed, otherwise it is false.
- *fogProcess*: A logical value (true–false), that is true, if the data must be processed in a fog node, and should not be sent to the cloud. It is generally set to true, when real-time response is needed from the fog node.
- *startTime*: The exact time in the simulator, when the data was generated.
- *processTime*: The exact time in the simulator, when the data was processed.
- *endTime*: The exact time in the simulator, when the response has been received by the actuator.
- *maxToleratedDelay* and *priorityLevel*: These two attributes define the maximum delay tolerated by the smart device and the priority of the data. Both of them could play a major role in task-scheduling algorithms (*e.g.* priority task scheduling), but they have no significant role in the current extension.
- *actuatorEvent*: This is the specific event type that is sent back to the actuator for execution.

To set these values accurately, some sensor-specific and environment-specific properties are required. The *SensorCharacteristics* class integrates these properties and helps to create more realistic simulations. The following attributes can be set:

- *sensorNum*: The number of applied sensors in a device. It is directly proportional to the size of the generated data.

- *mttf*: The mean time until the sensor fails. This attribute is essential to calculate the sensor's average life expectancy, which helps in modelling sensor failure events. If the simulation's time exceeds the *mttf* value, the sensor has a higher chance to fail. If a sensor fails, the actuator forces it to stop.

- *minFreq* and *maxFreq*: These two numbers represent the maximum and minimum sampling rate of the sensor. If a sensor does not have a predefined sampling rate but rather senses changes in the environment, then these are environment-specific attributes and their values could be defined by estimating the minimum and maximum time interval between state changes in the environment. These attributes are necessary to limit the possible frequency value of a sensor when the actuator imposes an event which affects the frequency.

- *fogDataRatio*: An estimation on how often the sensor generates data, that requires a fog process. This value is usually higher in the case of sensors that generate sensitive data or applications that require real-time response.

- *actuatorRatio*: An estimation on how often the sensor generates data, that requires actuator action. This is typically an environment-specific attribute. The more inconsistent and variable the environment is, the higher its chance to trigger the actuator, thus the value of this attribute should be set higher. This attribute has an impact on the *DataCapsule*'s *actuationNeeded* value. If the *actuatorRatio* is higher, then it is more likely to set the *actuationNeeded* attribute to true.

- *maxLatency*: Its value determines the maximum latency tolerated by the device when communicating with a computing node. For instance, in the case of medical devices, this value is generally lower than in the case of agricultural sensors. Mobile devices may move away from fog nodes inducing latency fluctuations and this attribute helps to determine whether a computing node is suitable for the device, or the expected latency exceeds this *maxLatency* limitation, therefore the device should look for a new computing node. This attribute plays a major role in triggering fog-selection actuator events when the IoT device is moving between fog nodes.

- *delay*: The delay of the data generating mechanism of the sensor.

When creating an IoT device in the simulation, its *SensorCharacteristics* features should also be defined. This will enable the simulated device to start generating *DataCapsule* objects according to its characteristics. This is the start of the life-cycle of a *DataCapsule* object. *DataCapsule* objects are forwarded to a certain *Application* of a fog node, based on the fog selection strategy of the device.

The corresponding timed method (called (tick())) of the *Application* is responsible for processing the data on a fog node. If the actual fog node has adequate resources to process the received data, the processing happens, and if the *actuationNeeded* attribute of the processed *DataCapsule* object was true, then it is sent back to the actuator (*i.e.* the data source) expanded with a specific *actuatorEvent* object (denoting an action to be performed by the actuator). After the data object stored in the *DataCapsule* is received by the actuator, it executes the event. If the *actuationNeeded* attribute was false during the

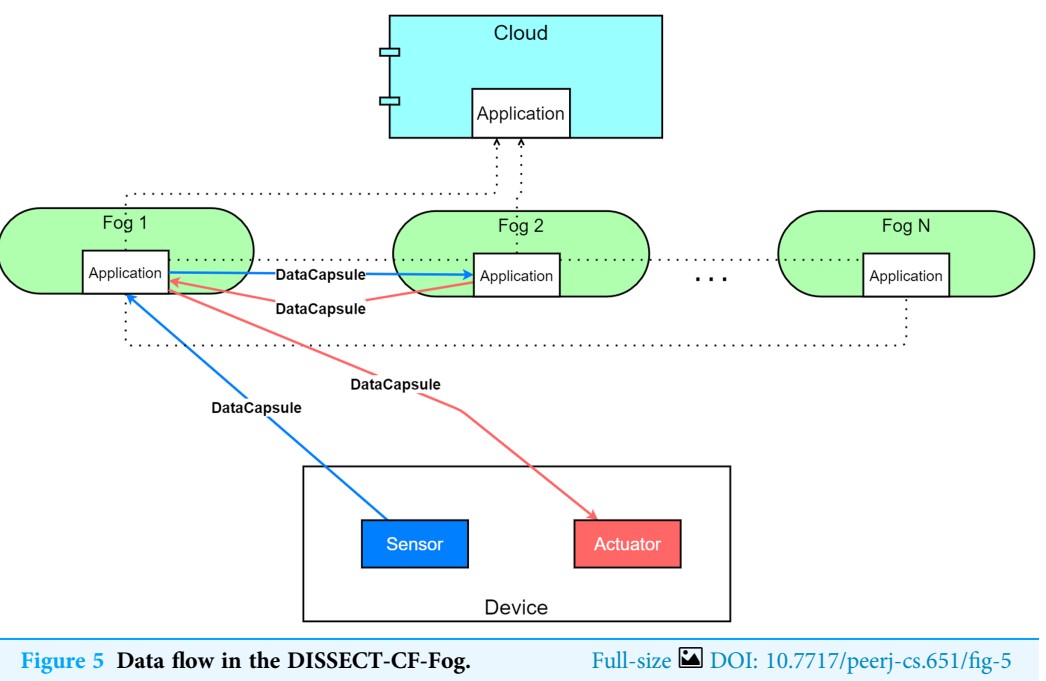

**Figure 5  Data flow in the DISSECT-CF-Fog.**

data processing, then the procedure mentioned before is omitted. Otherwise, if the current fog node does not have the capacity to process the data, it sends them over to another node based on the actual strategy of the *Application*.

The life-cycle of a *DataCapsule* object ends, when the actuator interface receives a notification indirectly by triggering a consumption event for which the IoT device has been subscribed to. By definition of the *DataCapsule*, its life-cycle can also end without actualisation events, if the *actuationNeeded* is set to false. A simplified demonstration of the *DataCapsule*'s path in the system (meaning the data flow) can be seen in Fig. 5.

The actuator model in DISSECT-CF-Fog is represented as the composition of three entities that highly depend on each other. These entities are the *Actuator*, *ActuatorStrategy* and the *ActuatorEvent*. The entities are serving input directly or indirectly for each other, as shown in Fig. 6.

As mentioned in "Actuator Model" the actuator model must only operate with predefined events to limit its scope to certain actions. These events are represented by the *ActuatorEvent* component, which is the core element of this model. By itself, the *ActuatorEvent* is only an interface and should be implemented in order to specify an exact action. There are some predefined events in the system: five of them are low-level, sensor-related events (as discussed in "Actuator Model"), the other five are related to the mobile functionality of the devices, but these can be extended to different types of behaviours.

Since the actuator has the ability to control the sensing process itself (*Pisani et al., 2020*), half of the predefined actuator events foster low-level sensor interactions. The *Change filesize* event can modify the size of the data to be generated by the sensor. Such behaviour reflects use cases, when more or less detailed data are required for the corresponding IoT application, or the data should be encrypted or compressed for some reason.

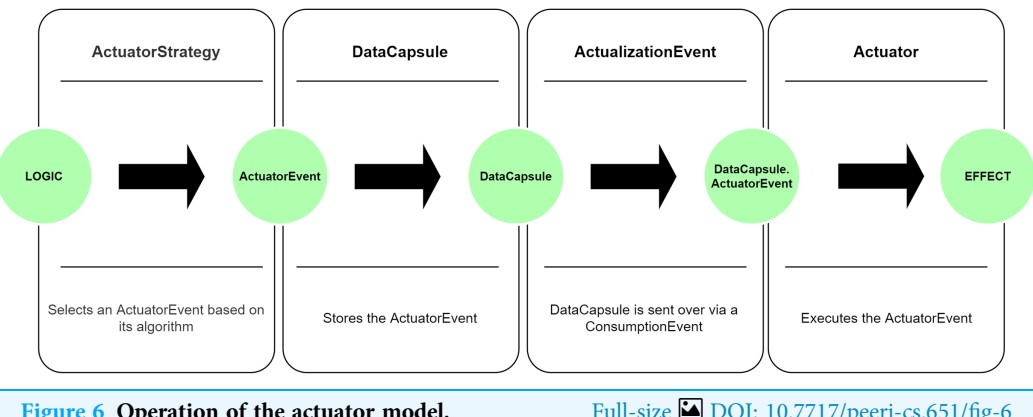

**Figure 6  Operation of the actuator model.**   

The *Increase frequency* and *Decrease frequency* might be useful when the IoT application requires an increased time interval between the measurements of a sensor. A typical use case of this behaviour is when a smart traffic control system of a smart city monitors the traffic at night, when usually less inhabitants are located outside. The maximum value of the frequency is regulated by the corresponding *SensorCharacteristics* object. The *Decrease frequency* is the opposite of the previously mentioned one, a typical procedure may appear in IoT healthcare, for instance the blood pressure sensor of a patient measures continuously increasing values, thus more frequent perceptions are required. The minimum value of the frequency is regulated by the corresponding *SensorCharacteristics*. The *Stop device* event imposes fatal error of a device, typically occurring randomly, and it is strongly related to the *mttf* of the *SensorCharacteristics*. The *mttf* is considered as a threshold, before reaching it, there is only a small chance for failure, after exceeding it, the chance of a failure increases exponentially. Finally, the *Restart device* reboots the given device to simulate software errors or updates.

Customised events can be added to the simulation by defining the *actuate()* method of the *ActuatorEvent* class, that describes the series of actions to occur upon executing the event. The event is selected corresponding to the *ActuatorStrategy*, which is a separate and reusable logic component and indispensable according to "Actuator Model". It is also an interface, and should be implemented to define scenario-specific behaviour. Despite its name, the *ActuatorStrategy* is capable of more than just simulating the configuration of an actuator and its event selection mechanism. This logic component can also be used to model the environmental changes and their side effects.

DISSECT-CF-Fog is a general fog simulator that is capable of simulating a broad spectrum of scenarios only by defining the key features and functionalities of each element of a fog and cloud infrastructure. The *ActuatorStrategy* makes it possible to represent an environment around an IoT device, and make the actuator component reactive to its changes. For instance, let us consider a humidity sensor and a possible implementation of the actuator component. We can then mimic an agricultural environment in the *ActuatorStrategy* with the help of some well-defined conditions to react to changes in humidity values, and select the appropriate customised actuator events (*e.g.* opening windows, or watering), accordingly. This characteristic enables DISSECT-CF-Fog to

simulate environment-specific scenarios, while maintaining its extensive and generic feature.

Finally, the *Actuator* component executes the implemented actions and events. There are two possible event executions offered by this object:

1. It can execute an event selected by the strategy. This is the typical usage, and it is performed automatically for devices needing actualisation, every time after the data have been processed by a computing unit, and a notification is sent back to the device.
2. Single events can also be fired by the actuator itself. If there is no need for an intermediate computing unit (*i.e.* data processing and reaction for the result), the actuator can act immediately, wherever it is needed as we mentioned in "Actuator Model".

There might be a delay between receiving an *ActuatorEvent* and actually executing it, especially when the execution of the event is a time consuming procedure. This possible delay can be set by the *latency* attribute of the *Actuator*. By default, a device has no inherent actuator component, but it can be explicitly set by the *setActuator()* method in order to fulfil the optional presence of the actuator as mentioned in "Actuator Model".

## Representing IoMT environments in DISSECT-CF-Fog

The basis of mobility implementations in the competing tools usually represent the position of users or devices as two or three-dimensional coordinate points, and the distance between any two points is calculated by the Euclidean distance, whereby the results can be slightly inaccurate. To overcome this issue and have a precise model (as we stated in "Requirements for modelling the Internet of Mobile Things"), we take into account the physical position of the end users, IoT devices and data centres (fog, cloud) by longitude and latitude values. The representative class called *GeoLocation* calculates distance using the Haversine formula. Furthermore, applying geographical location with a coordinate system often results in a restricted map, where the entities are able to move, thus in our case worldwide use cases can be implemented and modelled.

In real life, the motion of an entity can be represented by a continuous function, however in DISSECT-CF-Fog the discrete events reflect the state of the function describing a motion, thus continuous movements are transformed into such events, for instance modifying the direction in discrete moments. Therefore, the actual position only matters and is evaluated before the decisions are made by a computing appliance or a device, for instance when the sensed data is ready to be forwarded.

As we stated in "Requirements for modelling the Internet of Mobile Things", the mobile device movements are based on certain strategies. Currently two mobility strategies are implemented. We decided to implement one entity and one group mobility model according to *Camp, Boleng & Davies (2002)*, but since we provide a mobility interface, the collection of the usable mobility models can be easily extended.

The goal of the (i) *Nomadic* mobility model is that entities move together from one location to another, in our realisation multiple locations (*i.e.* targets) are available. It is very similar to the public transport of a city, where the route can be described by predefined

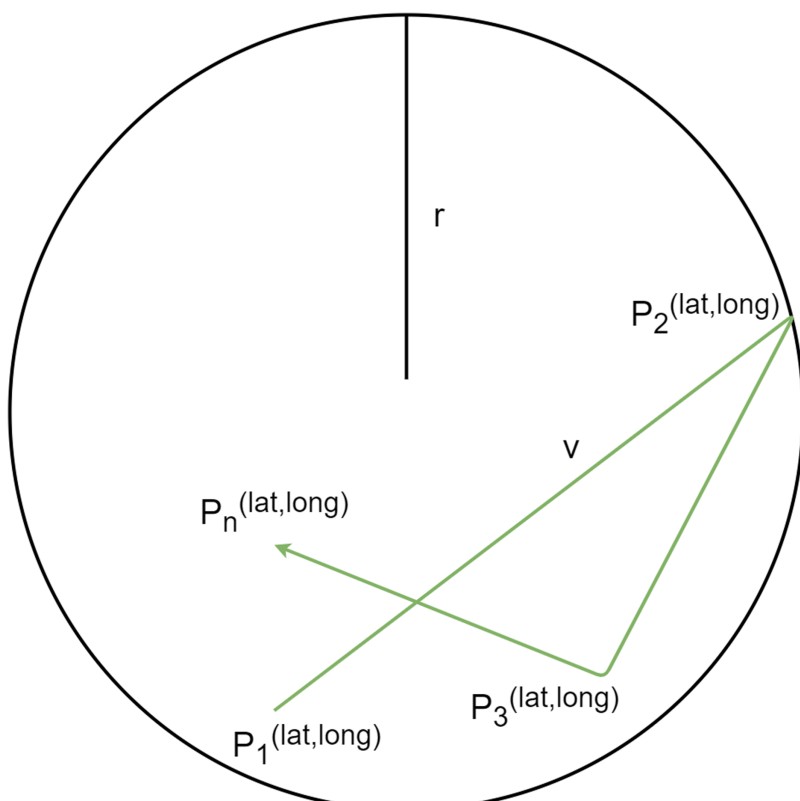

**Figure 7 Random Walk mobility model.**

points (or bus stops), and the dedicated points ($Pi$) are defined as entities of the *GeoLocation* class. An entity reaching the final point of the route will no longer move, but may function afterwards. Between the locations, a constant $v$ speed is considered, and there is a fixed order of the stops as follows:

$$P_1^{(lat,long)} \xrightarrow{v} P_2^{(lat,long)} \xrightarrow{v} \ldots \xrightarrow{v} P_n^{(lat,long)}$$

The (ii) *Random Walk* mobility takes into consideration entities with unexpected and unforeseen movements, for instance the observed entity walks around the city, unpredictably. The aim of this policy is to avoid moving in straight lines with a constant speed during the simulation, because such movements are unrealistic. In this policy, a range of the entity is fixed ($r$), where it can move with a random speed ($v$). From time to time, or if the entity reaches the border of the range, the direction and the speed of the movement dynamically change ($Pi$). That kind of movement is illustrated in Fig. 7.

The *MobilityDecisionMaker* class is responsible for monitoring the position of the fog nodes and IoT devices, and making decisions knowing these properties. This class has two main methods. The (i) *handleDisconnectFromNode()* closes the connection with the corresponding node in case the latency exceeds the maximum tolerable limit of the device, or the IoT device is located outside of the range of the node. The (ii) *handleConnectToNode ()* method is used, when a device finds a better fog node instead of the current one, or

the IoT device runs without connection to any node, and it finds an appropriate one. These methods are directly using the actuator interface to execute the corresponding mobility-based actuator events.

As we mentioned earlier, actuation and mobility are interlinked, thus we introduce five actuator events related to mobility according to "Requirements for modelling the Internet of Mobile Things". Position changes are done by *Change position* event of the actuator. The connection or disconnection methods of a device are handled by the *Disconnect from node* and *Connect to node* events, respectively. When a more suitable node is available for a device than the already connected one, the *Change node* actuator event is called. Finally, in some cases a node may stay without any connection options due to its position, or in cases when only overloaded or badly equipped fog nodes are located in its neighbourhood. The *Timeout* event is used to measure the unprocessed data due to these conditions, and to empty the device's local repository, if data forwarding is not possible.

## EVALUATION

We evaluated the proposed actuator and mobility extensions of the DISSECT-CF-Fog simulator with two different scenarios, which belong to the main open research challenges in the IoT field (*Marjani et al., 2017*). The goal of these scenarios is to present the usability and broad applicability of our proposed simulation extension. We also extended one of the scenarios with larger scale experiments, in order to determine the limitations of DISSECT-CF-Fog (*e.g.* determining the possible maximum number of simulated entities).

Our first scenario is IoT-assisted logistics, where more precise location tracking of products and trucks can be realised, than with traditional methods. It can be useful for route planning (*e.g.* for avoiding traffic jams or reducing fuel consumption), or for better coping with different environmental conditions (*e.g.* for making weather-specific decisions).

Our second scenario is IoT-assisted (or smart) healthcare, where both monitoring and reporting abilities of the smart systems are heavily relied on. Sensors wore by patients continuously monitor the health state of the observed people, and in case of data spikes it can immediately alarm the corresponding nurses or doctors.

During the evaluation of our simulator extension we envisaged a distributed computing infrastructure composed of a certain number of fog nodes (hired from local fog providers) to serve the computational needs of our IoT applications. Beside these fog resources, additional cloud resources can be hired from a public cloud provider. For each of the experiments, we used the cloud schema of LPDS Cloud of MTA SZTAKI[1] to determine realistic CPU processing power and memory usage for the physical machines. Based on this schema we attached 24 CPU cores and 112 GB of memory for a fog node, and set at most 48 CPU cores and 196 GB of memory to be hired from a cloud provider to start virtual machines (VMs) for additional data processing.

The simulator can also calculate resource usage costs, so we set VM prices according to the Amazon Web Services[2] (AWS) public cloud pricing scheme. For a cloud VM having 8 CPU cores and 16 GB RAMs we set 0.204$ hourly price (*a1.2xlarge*), while for a fog

[1] LPDS Cloud of the MTA SZTAKI website is available at: https://www.sztaki.hu/en/science/departments/lpds (accessed in October, 2020).

[2] The Amazon Web Service website is available at: https://aws.amazon.com/ec2/pricing/on-demand/ (accessed in October, 2020).

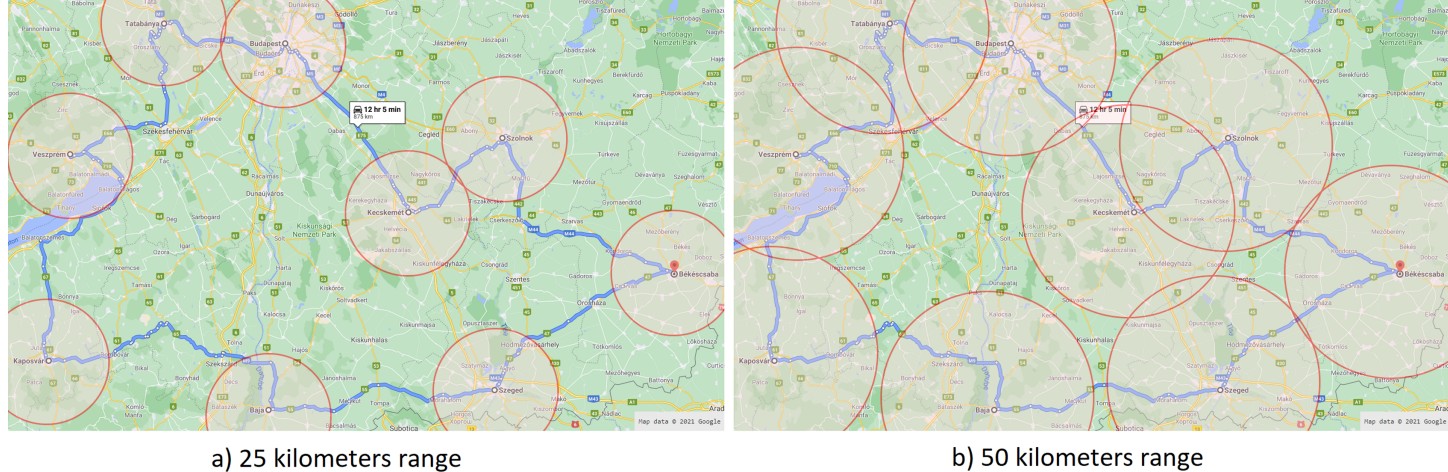

a) 25 kilometers range                    b) 50 kilometers range

**Figure 8 Applied fog ranges in the first scenario.**               

VM having four CPU cores and 8 GB RAMs we set 0.102$ hourly price (*a1.xlarge*). This means that the same amount of data is processed twice faster on the stronger, cloud VM, however the cloud provider also charges twice as much money for it. In our experiments, we proportionally scale the processing time of data, for every 50 kBytes, we model one minute of processing time on the Cloud VM.

For both scenarios, we used a PC with Intel Core i5-4460 3.2 GHz, 8 GB RAM and a 64-bit Windows 10 operating system to run the simulations. Since our simulations take into account random factors, each experiment was executed ten times, and the average values are presented below.

### The logistics IoT scenario

In the first scenario, we simulated a one year long operation of a smart transport route across cities located in Hungary. This track is exactly 875 km long, and it takes slightly more than 12 h to drive through it by a car based on the Google Maps, which means the average speed of a vehicle is about 73 km/h.

We placed fog nodes in nine different cities maintained by a domestic company, and we used a single cloud node of a cloud provider located in Frankfurt. Each fog node has direct connection with the cloud node, the latency between them is set based on the values provided by the WonderNetwork service[3]. A fog node forms a cluster with the subsequent and the previous fog node on the route as depicted in Fig. 8. This figure also presents the first test case (a), when the range of a fog node is considered as 25 km radius (similarly to a LoRa network). For the second test case (b), we doubled the range to 50 km radius. The IoT devices (placed in the vehicles to be monitored) were modelled with 4G network options with an average 50 ms of latency.

All vehicles were equipped by three sensors (asset tracking sensor, AIDC (automatic identification and data capture) and RFID (radio-frequency identification)) generating 150 bytes[4] of data per sensor. A daemon service on the computational node checks the

[3] The WonderNetwork website is available at: https://wondernetwork.com/pings (accessed in October, 2020).

[4] The Ericsson website is available at: https://www.ericsson.com/en/mobility-report/articles/massive-iot-in-the-city (accessed in May, 2021).

**Table 3** Results of the random actuator strategy and number of events during the first scenario.

| Actuator strategy | Random | | | | | |
|---|---|---|---|---|---|---|
| Fog node range (km) | 25 | | | 50 | | |
| Vehicle (pc.) | 2 | 20 | 200 | 2 | 20 | 200 |
| VM (pc.) | 19 | 19 | 19 | 19 | 19 | 19 |
| Generated data (MB) | 48 | 491 | 4,868 | 79 | 801 | 8,025 |
| Fog + Cloud cost ($) | 1,988.5 | 2,973.1 | 9,619.9 | 3061.1 | 4,026.1 | 10,357.4 |
| Delay (min.) | 5.0 | 4.01 | 2.03 | 5.0 | 4.02 | 2.02 |
| Runtime (sec.) | 3 | 13 | 141 | 4 | 16 | 169 |
| Change file size (pc.) | 20,937 | 210,009 | 2,102,215 | 34,873 | 348,226 | 3,477,983 |
| Change node (pc.) | 0 | 0 | 0 | 11,573 | 115,535 | 1,155,243 |
| Change position (pc.) | 181,388 | 1,812,784 | 18,137,172 | 181,447 | 1,814,159 | 18,141,355 |
| Connect/disconnect to node (pc.) | 12,985 | 129,944 | 1,299,099 | 1,556 | 15,833 | 158,751 |
| Increase frequency (pc.) | 21,239 | 210,352 | 2,104,912 | 34,812 | 346,774 | 3,479,261 |
| Decrease frequency (pc.) | 10,591 | 105,888 | 1,059,124 | 17,282 | 174,314 | 1,739,929 |
| Restart/stop device (pc.) | 0 | 0 | 0 | 0 | 0 | 0 |
| Timeout (pc.) | 70,941 | 709,384 | 7,091,262 | 0 | 0 | 0 |
| Timeout data (MB) | 27 | 274 | 2,752 | 0 | 0 | 0 |

local storage for unprocessed data in every five minutes, and allocates them in a VM for processing. Each simulation run deals with increasing number of IoT entities, we initialise 2, 20 and 200 vehicles in every twelve hours, which go around on the route. Half of the created objects are intended to start their movements in the opposite direction (selected randomly).

During our experiments, we considered two different actuator strategies: the (i) *RandomEvent* models a chaotic system behaviour, where both mobility and randomly appearing actualisation events of a sensor can happen. The failure rate of IoT components *mttf* were set to 90% of a year, and avoiding unrealistically low or high data generation frequencies, we limited them to a range of one to 15 min (*minFreq,maxFreq*). Finally, we enhanced the unpredictability of the system by setting the *actuatorRatio* to 50%. The (ii) *TransportEvent* actuator policy defines a more realistic strategy to model asset tracking, which aims to follow objects based on a broadcasting technology (*e.g.* GPS). A typical use case of this, when a warehouse can prepare for receiving supplies according to the actual location of the truck. In our evaluation, if the asset was located closer than 5 km, it would send position data in every 2 min. In case of 5 to 10 km, the data frequency is 5 min, and from 10 to 30, the data generation is set to 10 min, lastly if it is farther than 30 km, it informs changes in 15 min.

The results are shown in Tables 3 and 4. The comparison are based on the following parameters: (i) *VM* reflects the number of created VMs during the simulation on the cloud and fog nodes, which process the amount of generated data. As we mentioned earlier, our simulation tool is able to calculate the utilisation cost of the resources based on the predefined pricing schemes (*Fog+Cloud cost*). *Delay* reflects the timespan between the

**Table 4 Results of the transport actuator strategy and number of events during the first scenario.**

| Actuator strategy | Transport | | | | | |
|---|---|---|---|---|---|---|
| Fog node range (km) | 25 | | | 50 | | |
| Vehicle (pc.) | 2 | 20 | 200 | 2 | 20 | 200 |
| VM (pc.) | 19 | 19 | 19 | 19 | 19 | 19 |
| Generated data (MB) | 65 | 642 | 6445 | 83 | 851 | 8469 |
| Fog + Cloud cost ($) | 1,974.7 | 4,492.9 | 10,231.1 | 2,557.8 | 5,006.5 | 10,312.7 |
| Delay (min.) | 5.0 | 4.03 | 2.02 | 5.0 | 4.04 | 4.01 |
| Runtime (sec.) | 3 | 13 | 119 | 4 | 15 | 128 |
| Change file size (pc.) | 20,012 | 198,221 | 1,986,157 | 20,107 | 189,693 | 1,870,594 |
| Change node (pc.) | 0 | 0 | 0 | 6,111 | 65,424 | 654,135 |
| Change position (pc.) | 91,167 | 910,014 | 9,122,057 | 93,088 | 970,373 | 9,791,859 |
| Connect/disconnect to node (pc.) | 13,140 | 131,455 | 1,314,037 | 7,029 | 66,349 | 659,573 |
| Increase frequency (pc.) | 19,833 | 198,888 | 1,982,648 | 19,573 | 66,117 | 1,872,881 |
| Decrease frequency (pc.) | 19,735 | 199,759 | 1,983,997 | 19,646 | 189,298 | 1,875,489 |
| Restart/stop device (pc.) | 0 | 0 | 0 | 0 | 0 | 0 |
| Timeout (pc.) | 35,379 | 354,788 | 3,536,881 | 0 | 0 | 0 |
| Timeout data (MB) | 15 | 149 | 1,557 | 0 | 0 | 0 |

time of last produced data and the last VM operation. *Runtime* is a metric describing how long the simulation run on the corresponding PC. The rest of the parameters are previously known, it shows the number of the defined actuator and mobility events. Nevertheless *Timeout data* is highlighting the amount of data lost, which could not been forwarded to any node, because the actual position of a vehicle is to far for all available nodes.

Interpreting the results, we can observe that in case of the 25 km range, the *RandomEvent* drops more than half (around 56.19%) of the unprocessed data losing information, whilst the same average is about 23.4% for the *TransportEvent*. In case of 50 km range, there is no data dropped, because the nodes roughly cover the route and the size of gaps cannot trigger the *Timeout* event. In contrary, the ranges do not cover each other in case of the 25 km range, which results in zero *Change node* event.

Based on the *Fog+Cloud cost* metric, one can observe that the *TransportEvent* utilises the cloud and fog resources more, than the *RandomEvent*, nevertheless the average price of a device (applying two vehicles) is about 1197.7$, in case of 20 assets it decreases to about 206.2$, and lastly operating 200 objects reduce the price to about 50.6$, which means that the continuous load of the vehicles utilises the VMs more effectively.

Since the IoT application frequency was set to five minutes, we considered the *Delay* acceptable, when it was equal or less than five minutes. Based on the results, all test cases fulfilled our expectation. It is worth mentioning that *mttf* might be effective only in simulating years of operation, thus neither software nor hardware error is triggered (*Restart/stop device*) in this case. The *Runtime* metric also points to the usability and reliability performance of DISSECT-CF-Fog; less than three minutes was required to

evaluate a one year long scenario with thousand of entities (*i.e.* simulated IoT devices and sensors running for a year).

## Smart healthcare scenario

In the second scenario, we continued our experiments with a smart healthcare case study. In this scenario, patients wear blood pressure and heart rate monitors. We automatically adjust the data sampling period if the monitors report off nominal behaviour: (i) in case of blood pressure lower than 90 or higher than 140; (ii) in case of heart rate values lower than 60, and higher than 100.

In this scenario, each patient represents a different data flow (starting from its IoT device), similarly to the previously mentioned way. First the data is forwarded to the fog layer, if the data processing is impossible there due to overloaded resources, then the data is moved to the cloud layer to be allocated to a VM for processing. As IoT healthcare requires as low latency as possible, the frequency of the daemon services on the computational node was set to one minute. Similarly to the first scenario, one measurement of a sensor creates (a message of) 150 bytes.

We focus on the maximum number of IoT devices which can be served with minimal latency by the available fog nodes, and we are also interested in the maximum tolerable delay, if the raw data is processed in the cloud. We applied the same VM parameters as in the previous scenario, and the simulation period took one day. We did not implement mobility in this scenario, nevertheless actualisation events were still required in case of health emergency to see how the system adapts to the unforeseen data.

Similarly to the first scenario, the hospital was assumed to use a public cloud node in Frankfurt, but it was also assumed to maintain three fog nodes on the premises of the hospital. During our experiments, we considered various number of patients (100, 1,000 and 10,000), and we investigated how the operating costs and delay change and adapt to the different the number of fog VMs and actualisation events.

Since each fog node is available in the local region, the communication latency was set randomly between 10 and 20 ms (regarding to AWS[5]), furthermore the *actuatorRatio* was set to 100%, because of the vital information of the sensed data, thus each measurement required some kind of actuation. The rest of the parameters were the same we used in the logistics scenario.

Our findings are depicted in Table 5. One can observe that the increasing number of applied fog nodes reduces the average costs per patient, in case of three fog nodes the mean cost (projected on one patient) is around 83.7$. This amount of money is continuously grows as the fog nodes are omitted one by one, the corresponding average operating costs are about 97.7$, 118.7$ and 124.0$, respectively, which means maintaining fog nodes also might be economically worthy.

Figure 9 presents the delay of the IoT application concerning the number of utilised fog and cloud nodes. Using a higher number of fog nodes can foster faster data processing, however in case of 10,000 patients, the best delay is 7.74 min, which points out that the utilised resources were overloaded. In the other cases the system managed the patients'

[5] The AWS Architecture Guidelines and Decisions website is available at: https://aws.amazon.com/blogs/compute/low-latency-computing-with-aws-local-zones-part-1/ (accessed in May, 2021).

**Table 5 Results and number of events during the second scenario.**

| Actuator strategy | Healthcare | | | | | | | | | | | |
|---|---|---|---|---|---|---|---|---|---|---|---|---|
| Fog/cloud node ratio | 3/1 | | | 2/1 | | | 1/1 | | | 0/1 | | |
| Patient (pc.) | 10,000 | 1,000 | 100 | 10,000 | 1,000 | 100 | 10,000 | 1,000 | 100 | 10,000 | 1,000 | 100 |
| VM (pc.) | 21 | 11 | 12 | 17 | 8 | 9 | 12 | 5 | 5 | 6 | 2 | 2 |
| Generated data (MB) | 251 | 27 | 2 | 231 | 27 | 2 | 197 | 27 | 2 | 145 | 27 | 2 |
| Fog + Cloud cost ($) | 48.1 | 25.1 | 27.8 | 42.0 | 19.7 | 22.7 | 35.2 | 15.3 | 14.9 | 37.1 | 10.8 | 9.9 |
| Delay (min.) | 7.74 | 1.41 | 1.06 | 9.51 | 1.46 | 1.07 | 9.50 | 1.79 | 1.05 | 14.8 | 2.44 | 1.22 |
| Runtime (sec.) | 8 | 1 | 1 | 8 | 1 | 1 | 8 | 1 | 1 | 11 | 1 | 1 |
| Increase frequency (pc.) | 132,125 | 14,687 | 1,431 | 119,954 | 14,192 | 1392 | 98,975 | 14,127 | 1,468 | 71,153 | 13,927 | 1,399 |
| Decrease frequency (pc.) | 750,751 | 80,718 | 8,068 | 684,829 | 80,845 | 8104 | 563,198 | 80,295 | 8,023 | 406,155 | 81,105 | 8,115 |
| Restart/stop device (pc.) | 0 | 0 | 0 | 0 | 0 | 0 | 0 | 0 | 0 | 0 | 0 | 0 |

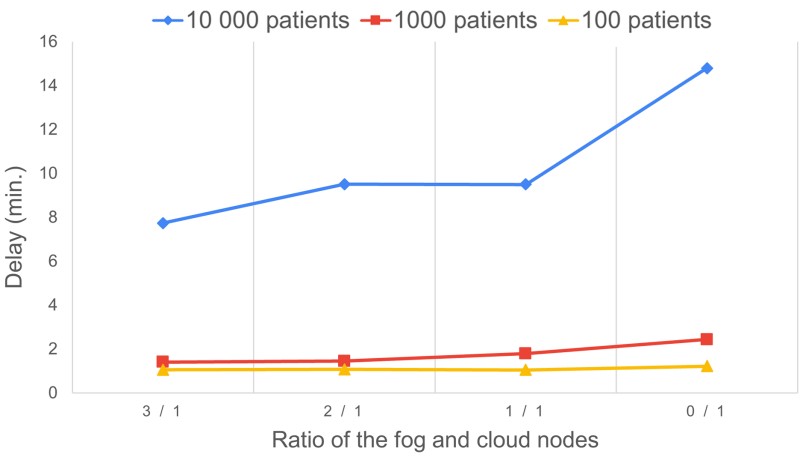

**Figure 9 Delay values of the second scenario.**

data with less than three minutes delay, but decreasing the number of usable fog nodes can continuously increase the delay.

Lastly, we can observe that no failure happened during the evaluation (*Restart/stop device*), because of the reliability of medical sensors and the short time of simulation. We can also realise that our simulation tool is able to model thousands of smart objects (*e.g.* IoT devices and sensors), and their one day long simulated operation could be done in 11 s of elapsed time (*Runtime*) in the worst case.

### Large-scale experiments of the smart healthcare scenario

In this section our goal was to point out the possible limitations of DISSECT-CF-Fog using the previously detailed smart healthcare scenario. The runtime of DISSECT-CF-Fog largely depends on the used execution environment and its actual hardware resources (mostly memory), similarly to any other software.

Our findings are presented in Table 6, in which we used the same metrics as before.

**Table 6 Results and number of events in the scalability studies.**

| Actuator strategy | Healthcare | | | | |
|---|---|---|---|---|---|
| Fog/cloud node ratio | 3/1 | | | 7/1 | 55/1 |
| Patient (pc.) | 170,000 | 180,000 | 190,000 | 190,000 | 190,000 |
| VM (pc.) | 24 | 24 | Out of memory | 48 | 336 |
| Generated data (MB) | 1,196 | 1,261 | | 1,513 | 1,679 |
| Fog + cloud cost ($) | 197.6 | 208.5 | | 244.9 | 674.9 |
| Delay (min.) | 6,256.0 | 6,796.0 | | 5,886.0 | 9.9 |
| Runtime (sec.) | 186 | 256 | | 159 | 163 |
| Increase frequency (pc.) | 624,860 | 657,725 | | 790,999 | 810,153 |
| Decrease frequency (pc.) | 3,557,783 | 3,751,754 | | 4,498,906 | 4,049,325 |
| Restart/stop device (pc.) | 0 | 0 | 0 | 0 | 0 |

For this scalability study, we also applied the earlier used topology with three fog nodes and a cloud node. To determine the exact number of IoT devices that can be modelled by the simulator is not possible, because our system takes into account random factors. Nevertheless, we can give an estimate by scaling of the number of IoT devices, in our case the amount of active devices (*i.e.* patients).

In this evaluation we increased the number of patients with 10,000 for the test cases, and examined the memory usage of the execution environment. The results showed that even for cases of 170,000 and 180,000 IoT devices, the fog and cloud nodes can process the vast amount of data generated by the modelled IoT sensors, however the *Delay* value also increased dramatically to 6 256 min, in the first case, and 6,796 min, in the second case. It is worth mentioning that besides such a huge number of active entities, the *Runtime* values are below 5 min. When we simulated 190,000 IoT devices, the simulator consumed all of the memory of the underlying hardware.

In the fourth test case, we applied seven fog nodes. Our findings showed that the *Delay* value decreased spectacularly to 5,886 min, however it is far from what we experienced in the second scenario, therefore our further goal was to define how many computational resources (*i.e.* fog nodes) are required to decrease the *Delay* parameter below 10 min, similarly to what we expected in the second scenario.

We can clearly seen in the fifth test case that at least 55 fog nodes are required for 190,000 IoT devices to process and store their data. In this case, the *Delay* value is 9.9 min, but because of the higher number of computational nodes, both numbers of the utilised VMs (336 pieces) and these costs (674.9$) increased heavily. The Java representation of the fog and cloud nodes hardly differ, therefore we could reach similar results, if we increased the number of cloud nodes as well.

It can be clearly seen that the critical part of DISSECT-CF-Fog is the number of IoT devices utilising in the system, however if we also increase the number of the simulated computing resources (*i.e.* fog and cloud nodes), we can reach better scalability (*i.e.* the delay and simulation runtime would not grow). The reason for this is that the actual Java implementation of DISSECT-CF-Fog stores the references of model entities of the devices

and the unprocessed data. To conclude, the current DISSECT-CF-Fog extension is capable of simulating even up to 200 thousand system entities. Limitations are only imposed by the the hardware parameters utilised, and the wrongly (or extremely) chosen ratio of the number of IoT devices and computing nodes set for the experiments.

## CONCLUSION

In this paper, we introduced the extended version of DISSECT-CF-Fog to support actuators and mobility features. Concerning our main contribution, we designed and developed an actuator model that enables broad configuration possibilities for investigating IoT-Fog-Cloud systems. With our extensions, various IoT device behaviours and management policies can be defined and evaluated with ease in this simulator.

We also evaluated our proposal with two different case studies of frequently used IoT applications, and we extended the smart healthcare scenario with large-scale experiments to determine the limitations of our approach. These IoT scenarios utilise the predefined actuator events of the simulator. We also presented how to use different actuator strategies, in order to define specific application (and sensor/actuator) behaviour. In essence, our solution ensures a compact, generic and extendable interface for actuator events, which is unique among state-of-the-art simulators in the area.

Our future work will address more detailed and extended mobility models for migration and resource scaling decisions. We also plan to extend the actuator strategies to model various types and behaviour of IoT entities.

### Funding

This research was supported by the Hungarian Scientific Research Fund under the grant number OTKA FK 131793, by the Hungarian Government under the grant number EFOP-3.6.1-16-2016-00008, and by the UNKP-21-3 New National Excellence Program of the Ministry for Innovation and Technology from the source of the National Research, Development and Innovation Fund. The funders had no role in study design, data collection and analysis, decision to publish, or preparation of the manuscript.

### Grant Disclosures

The following grant information was disclosed by the authors:
Hungarian Scientific Research Fund: OTKA FK 131793.
Hungarian Government: EFOP-3.6.1-16-2016-00008.
National Research, Development and Innovation Fund: UNKP-21-3.

### Competing Interests

The authors declare that they have no competing interests.

### Author Contributions

- Andras Markus performed the experiments, analyzed the data, performed the computation work, prepared figures and/or tables, and approved the final draft.

- Mate Biro performed the experiments, performed the computation work, prepared figures and/or tables, and approved the final draft.
- Gabor Kecskemeti conceived and designed the experiments, performed the computation work, authored or reviewed drafts of the paper, and approved the final draft.
- Attila Kertesz conceived and designed the experiments, analyzed the data, authored or reviewed drafts of the paper, and approved the final draft.

## Data Availability

The source code of the extension is available at GitHub:

https://github.com/andrasmarkus/dissect-cf/tree/actuator.

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
