# Peer review of "Actuator behaviour modelling in IoT-Fog-Cloud simulation"

_PeerJ Computer Science, doi:10.7717/peerj-cs.651_

## Round 0.1 · original submission · Major Revisions

The work needs substantial rewriting and restructuring on several aspects to be accepted. Failure to perform such restructuring and improvement will result in a rejection.

In addressing reviewers' concerns pay attention especially to Reviewer 2's comments, which highlight important issues.

Reviewer 1 ·

Basic reporting

The paper well written and presented. The related works are considered. The paper structure and figures are adequate.

However, the paper do not clearly explain why the proposed simulator is better than previously implemented Edge/Fog simulators.

Experimental design

1. The validation methodology is not presented at all. This is essential for the simulation paper.

2. Why authors use Java to implement their simulator?!?! Java is a very bad choice for suimulators since it has a poor performance compared to C/C++.

Validity of the findings

Unfortunately, authors do not validate the accuracy of their simulator. It is impossible to check if the simulator works or not.

Additional comments

In this paper, authors present a new simulator, DISSECT-CF-Fog, to model operating costs and latencies for Cloud/Edge/Fog infrastructures. Even though the paper is well written and presented, I think the provided results and conclusions cannot be verified since authors do not provide any validation methodology or validation results for their simulator.

I understand that there are a lot of papers that present similar simulators without providing any justification for their accuracy. However, I think that such papers cannot be considered as scientific since the presented experimental results and conclusions are not verifiable (see works from Karl Popper).

Overall, I have the following major concerns:

1. There is no any validation/verification chapter which clearly explains how authors verified their simulator.

2. Authors do not clearly explain how DISSECT-CF-Fog differs from previous simulators. Why is it better?

3. I do not understand why authors used Java for a simulator!!! This is obviously a wrong choice given the fact that Java was not designed for high performance computing. Why authors did not use C/C++?

·

Basic reporting

See General comments for the author

Experimental design

See General comments for the author

Validity of the findings

See General comments for the author

Additional comments

This paper proposes an extended version of DISSECT-CF-Fog to support actuators and mobility and analyze their performances in terms of cost and execution time. It introduced a comparison to the existing simulators in terms of communication direction, actuator events, mobility, and position. Overall, the subject considered in the paper is topical and important. The paper is properly organized, and relatively easy to follow. The general idea of the proposed solution is clear, but the authors should justify the motivation for the proposed solution. On the downside, there are loose points in the paper in which authors should address them in a revised paper. My concerns are listed below:

- Please improve the quality of the paper presentation (e.g., in the introduction, try to further emphasize the relevance of this work).
- Please consider performing multiple repetitions of the experiments (e.g., to perform statistical hypothesis testing) and to use a more complex scenario.
- What about the scalability problem? Does the proposed model work for large cloud data center and under any offered workload? In other words, if the number of IoT devices and servers increases, what is the impact on the performance of the proposed model?
- In the second use case, what will happen if we exceed 10,000 patients? Does the system performance not deteriorate?
- According to existing research work, to have reliable results, it is necessary to perform each experiment ten times and we take the average values instead of the three times that you have mentioned in the evaluation Section.
- Please present the limitations of your work.

Minor comments:

- There are grammatical errors and typos throughout the entire manuscript. For instance :
o Add the figure number in the following sentence: "These kinds of actions are noted by gray dashed arrows in the figure".
o Correct the word “actuator” in Table 2.
- The English used in the paper is not bad. However, it will be important to revise the English well to further improve the paper.

---

## Round 0.2 · Major Revisions

In addition to provide the requested clarifications, please consider carefully the comments of reviewer 3, especially those related to the assessment.

·

Basic reporting

no comment

Experimental design

no comment

Validity of the findings

no comment

Additional comments

This version has well addressed all my concerns.

Reviewer 3 ·

Basic reporting

The paper provides an overview on an extension of prior work, the implementation of an "actuator" (subscriber) model for the standalone IoT/Fog simulator DISSECT-CF-Fog.

The storyline of the paper requires some improvements, to make the paper more objective (in the abstract, in introduction).

The abstract needs to be simplified and present concrete contributions and achieved results.

The terms used, sensor/actuator is misleading and should be revised. The authors address IoT environments, where there are, from an abstract perspective, publishers and subscribers. An actuator is a device that is used to manipulate the physical environment, e.g., temperature control.
While a consumer can be an actuator but also any other type of cyber-physical system. Or a software-based agent. If the authors truly want to model actuators, which type are they addressing? Linear, motors, relays, solenoids?

In related work, FogFlow is missing. Did the authors consider a comparison?
Also, the "mobility" feature should benefit from a more clear description. Which type of mobility is being covered? consumer? publisher? any node?

- A background section introducing DISSECT-CF-Fog should be added, as it would help to understand whether or not the contributions of this paper agains prior work are relevant.

- The implementation section currently provides a high-level specification of the actuator module. This should be changed to make the flow of the implementation clearer.

The decision on relying on random mobility (which has been shown not to be the best model in terms of simulations) needs to be justified.

Experimental design

- The experimental aspects described in section 4 needs to be made more objective, avoiding claims such as relying on two of the most frequently applied use-cases - do the authors have a proof on this?
- Several papers debate scenarios for Fog relying on different simulators. One of the problems with the experimental design are the design choices, such as selecting latency to be between 10 ms and 20ms due to fog node locality/co-placement. The authors should justify the choice of parameters, or propose parameters that can provide a better understanding of the variability impact, creating scenarios with values for "low", "medium", "high" delay, etc. Similar remarks go to other parameters, such as node cost.
- a patient, in the simulations, stands exactly for what? 1 data flow, VBR or CBR? What is the size of the packets sent? The overall load is given, but what is the impact in terms of patients served? Similarly for all parameters described in Table 5.
- for the large-scale experiments, where are the results?

Validity of the findings

- the value-add of the proposed extension is not clear, as there was no comparison against a specific benchmark.
- the experimental design impacts the overall validity, as the selection of parameters seems to have been done in an ad-hoc way.
- Not clear what is the value-add for mobile scenarios, and the introduction states that the extension has been developed for an Internet of Mobile things...

---

## Round 0.3 · accepted · Accept

Please perform careful proofreading as hinted by the reviewer.

Reviewer 4 ·

Basic reporting

English should be polished a little more...E.g.:
- data is a plural noun
- of DISSECT-CF-Fog strongly build -> of DISSECT-CF-Fog is strongly built
- .By -> . By

Other issues seem to be solved

Experimental design

the raised issues seem to be solved

Validity of the findings

the raised issues seem to be solved